# Beyond the Single Neuron Convex Barrier
# for Neural Network Certification

Gagandeep Singh[1], Rupanshu Ganvir[2], Markus Püschel[1], Martin Vechev[1]
Department of Computer Science
ETH Zurich, Switzerland
[1]{gsingh,pueschel,martin.vechev}@inf.ethz.ch
[2]rganvir@student.ethz.ch

## Abstract

We propose a new parametric framework, called k-ReLU, for computing precise and scalable convex relaxations used to certify neural networks. The key idea is to approximate the output of multiple ReLUs in a layer *jointly* instead of separately. This joint relaxation captures dependencies between the inputs to different ReLUs in a layer and thus overcomes the convex barrier imposed by the single neuron triangle relaxation and its approximations. The framework is parametric in the number of $k$ ReLUs it considers jointly and can be combined with existing verifiers in order to improve their precision. Our experimental results show that k-ReLU enables significantly more precise certification than existing state-of-the-art verifiers while maintaining scalability.

## 1   Introduction

Neural networks are being increasingly used in many safety critical domains including autonomous driving, medical devices, and face recognition. Thus, it is important to ensure they are provably robust and cannot be fooled by *adversarial examples* [1]: small changes to a given image that can fool the network into making a wrong classification. To address this challenge, a range of verification techniques were introduced recently ranging from exact but expensive methods based on SMT solvers [2–4], mixed integer linear programming [5], and Lipschitz optimization [6] to approximate and incomplete, but more scalable methods based on abstract interpretation [7–9], duality [10, 11], semi definite [12, 13] and linear relaxations [14–17]. Recently, combinations of approximate methods with solvers have been used for producing more precise results than approximate methods alone while also being more scalable than exact methods [18, 19].

The key challenge any verification method must address is computing the output of ReLU assignments where the input can take both positive and negative values. Exact computation must consider two paths per neuron, which quickly becomes infeasible due to a combinatorial explosion while approximate methods trade precision for scalability via a convex relaxation of ReLU outputs.

The most precise convex relaxation of ReLU output is based on the convex hull of Polyhedra [20] which is practically infeasible as it requires an exponential number of convex hull computations, each with a worst-case exponential complexity in the number of neurons. The most common convex relaxation of $y_1 \text{:=} \text{ReLU}(x_1)$ used in practice [17, 5] is the triangle relaxation from [3]. We note that other works such as [8, 9, 14–16, 11] approximate this relaxation. The triangle relaxation creates constraints only between $y_1$ and $x_1$ and is optimal in the $x_1 y_1$-plane. Because of this optimality, recent work [17] refers to the triangle relaxation as the convex barrier, meaning the best convex approximation one can obtain when processing each ReLU separately. However, the triangle relaxation is *not optimal* when one considers multiple neurons at a time as it ignores all dependencies between $x_1$ and any other neuron $x_2$ in the same layer, and thus loses precision.

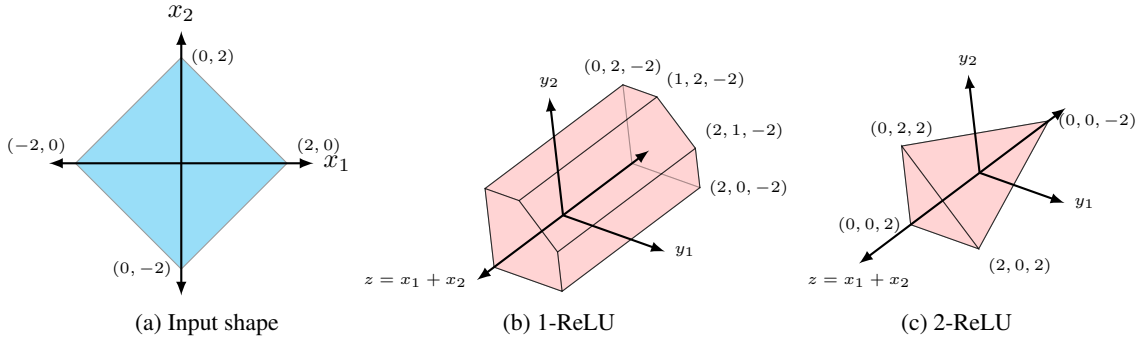

Figure 1: The input space for the ReLU assignments $y_1 := ReLU(x_1)$, $y_2 := ReLU(x_2)$ is shown on the left in blue. Shapes of the relaxations projected to 3D are shown on the right in red.

**This work: beyond the single neuron convex barrier** In this work, we address this issue by proposing a novel parameterized framework, called k-ReLU, for generating convex approximations that consider multiple ReLUs *jointly*. Here, the parameter $k$ determines how many ReLUs are considered jointly with large $k$ resulting in more precise output. For example, unlike prior work, our framework can generate a convex relaxation for $y_1$:=ReLU($x_1$) and $y_2$:=ReLU($x_2$) that is optimal in the $x_1 x_2 y_1 y_2$-space. We next illustrate this point with an example.

**Precision gain with k-ReLU on an example** Consider the input space of $x_1 x_2$ as defined by the blue area in Fig. 1 and the ReLU operations $y_1$:=ReLU($x_1$) and $y_2$:=ReLU($x_2$). The input space is bounded by the relational constraints $x_2 - x_1 \leq 2$, $x_1 - x_2 \leq 2$, $x_1 + x_2 \leq 2$ and $-x_1 - x_2 \leq 2$. The relaxations produced are in a four dimensional space of $x_1 x_2 y_1 y_2$. For simplicity of presentation, we show the feasible shape of $y_1 y_2$ as a function of $z = x_1 + x_2$.

The triangle relaxation from [3] is in fact a special case of our framework with $k = 1$, that is, 1-ReLU. 1-ReLU independently computes two relaxations - one in the $x_1 y_1$ space and the other in the $x_2 y_2$ space. The final relaxation is the cartesian product of the feasible sets of the two individually computed relaxations and is oblivious to any correlations between $x_1$ and $x_2$. The relaxation adds triangle constraints $\{y_1 \geq 0, y_1 \geq x_1, y_1 \leq 0.5 \cdot x_1 + 1\}$ between $x_1$ and $y_1$ as well as $\{y_2 \geq 0, y_2 \geq x_2, y_2 \leq 0.5 \cdot x_2 + 1\}$ between $x_2$ and $y_2$.

In contrast, 2-ReLU considers the two ReLU's *jointly* and captures the relational constraints between $x_1$ and $x_2$. 2-ReLU computes the following relaxation:

$$\{y_1 \geq 0, y_1 \geq x_1, y_2 \geq 0, y_2 \geq x_2, 2 \cdot y_1 + 2 \cdot y_2 - x_1 - x_2 \leq 2\}$$

The polytope produced is shown in Fig. 1c. Note that in this case the shape of $y_1 y_2$ is not independent of $x_1 + x_2$ as opposed to the triangle relaxation. At the same time, it is more precise than Fig. 1b for all values of $z$.

**Main contributions** Our main contributions are:

- A novel framework, called k-ReLU, that computes optimal convex relaxations for the output of $k$ ReLU operations *jointly*. k-ReLU is generic and can be combined with existing verifiers for improved precision while maintaining scalability. Further, k-ReLU is also adaptive and can be tuned to balance precision and scalability by varying $k$.

- A method for computing approximations of the optimal relaxations for larger $k$, which is more precise than simply using $l < k$.

- An instantiation of k-ReLU with the recent DeepPoly convex relaxation [9] resulting in a verifier called *kPoly*.

- An evaluation showing *kPoly* is more precise and scalable than the state-of-the-art verifiers [9, 19] on the task of certifying neural networks of up to 100K neurons against challenging adversarial perturbations (e.g., $L_\infty$ balls with $\epsilon = 0.3$).

We note that the work of [12] computes semi definite relaxations that consider multiple ReLUs jointly, however these are not optimal and do not scale to the large networks used in our experiments.

Table 1: Volume of the output bounding box computed by *kPoly* on a $9 \times 200$ network.

| $k$ | 1-ReLU | 2-ReLU | 3-ReLU |
|---|---|---|---|
| Volume | $4.5272 \cdot 10^{14}$ | $5.1252 \cdot 10^{7}$ | $2.9679 \cdot 10^{5}$ |

**Precision gain in practice** Table 1 quantitatively compares the precision of *kPoly* instantiated with three relaxations: $k = 1$, $k = 2$, and $k = 3$. We measure the volume of the output bounding box computed after propagating an $L_\infty$-ball of radius $\epsilon = 0.015$ through a deep, fully connected MNIST network with 9 layers containing 200 neurons each. We can observe that the volume of the output from 3-ReLU and 2-ReLU is respectively 9 and 7 orders of magnitude smaller than from 1-ReLU.

We note that the networks we consider, as for example the $9 \times 200$ network above, are especially challenging for state-of-the-art verifiers as these methods either lose unnecessary precision [8, 9, 14, 15, 19, 16, 17] or simply do not scale [5, 18, 12, 4, 13, 10].

Finally, we remark that while we consider robustness certification against norm based perturbations in our evaluation, our framework can also be used for precise and scalable verification of other network safety properties such as stability [21] or robustness against geometric and semantic perturbations [9, 22, 23].

## 2 Overview of k-ReLU

We now show, on a simple example, that the k-ReLU concept can be used to improve the results of state-of-the-art verifiers. In particular, we illustrate how the output of our verifier *kPoly* instantiated with 1-ReLU is refined by instantiating it with 2-ReLU. This is possible as the 2-ReLU relaxation can capture extra relationships between neurons that 1-ReLU inherently cannot.

Consider the simple feedforward neural network with ReLU activations shown in Fig. 2. The network has two inputs each taking values independently in the range $[-1, 1]$, one hidden layer and one output layer each containing two neurons. For simplicity, we split each layer into two parts: one for the affine transformation and the other for the ReLU. The weights of the affine transformation are shown on the arrows and the biases are above or below the respective neuron. The goal is to verify that $x_9 \leq 4$ holds for the output $x_9$ with respect to all inputs.

We first show that 1-ReLU instantiated with the state-of-the-art DeepPoly [9] relaxation fails to verify the property. DeepPoly, described formally in Section 4, associates two pairs of lower and upper bounds with each neuron $x_i$: $(a_i^{\leq}, a_i^{\geq})$ and $(l_i, u_i)$. Here, $a_i^{\leq}$ and $a_i^{\geq}$ have the form $\sum_j a_j \cdot x_j + c$ where $c, l_i, u_i, \in \mathbb{R} \cup \{-\infty, +\infty\}$ and $a_j \in \mathbb{R}$. The bounds computed by the verifier using this instantiation are shown as annotations in Fig. 2.

**First Layer** The verifier starts by computing the bounds for $x_1$ and $x_2$ which are simply taken from the input specification resulting in:

$$x_1 \geq -1, \ x_1 \leq 1, \ l_1 = -1, \ u_1 = 1, \text{ and } x_2 \geq -1, \ x_2 \leq 1, \ l_2 = -1, \ u_2 = 1.$$

**Second Layer** Next, the affine assignments $x_3 := x_1 + x_2$ and $x_4 := x_1 - x_2$ are handled. DeepPoly handles affine transformations exactly and thus no precision is lost. The affine transformation results in the following bounds for $x_3$ and $x_4$:

$$x_3 \geq x_1 + x_2, x_3 \leq x_1 + x_2, l_3 = -2, u_3 = 2,$$
$$x_4 \geq x_1 - x_2, x_4 \leq x_1 - x_2, l_4 = -2, u_4 = 2.$$

DeepPoly can precisely handle ReLU assignments when the input neuron takes only positive or negative values, otherwise it loses precision. Since $x_3$ and $x_4$ can take both positive and negative values, an approximation based on the triangle relaxation is applied which for $x_5$ yields:

$$x_5 \geq 0, \quad x_5 \leq 1 + 0.5 \cdot x_3. \tag{1}$$

Note that DeepPoly discards the other lower bound $x_5 \geq x_3$ from the triangle relaxation. The lower bound $l_5$ is set to 0 and the relation $x_3 \leq x_1 + x_2$ is substituted for $x_3$ in (1) for computing the upper bound which yields $l_5 = 0, u_5 = 2$. Analogously, for $x_6$ we obtain:

$$x_6 \geq 0, \quad x_6 \leq 1 + 0.5 \cdot x_4, \quad l_6 = 0, \quad u_6 = 2. \tag{2}$$

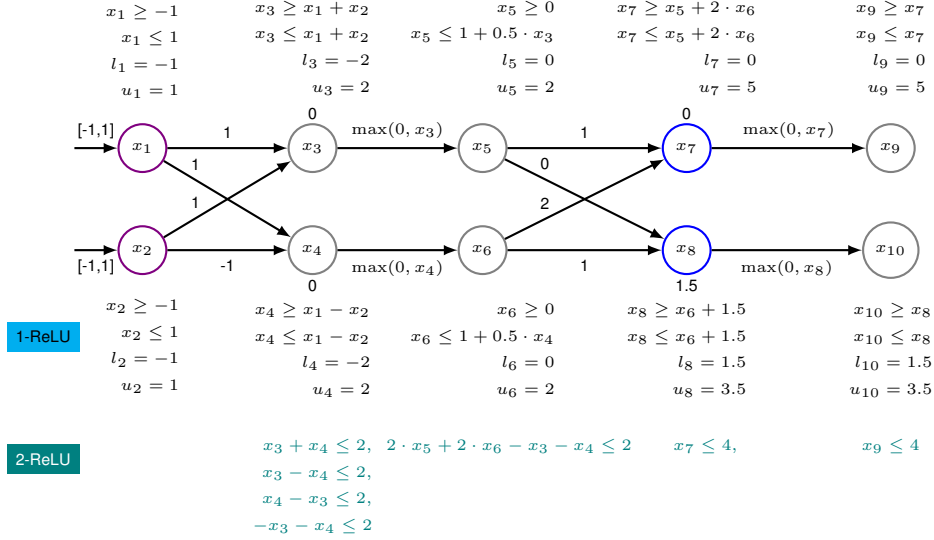

Figure 2: Verification of property $x_9 \leq 2$. Refining DeepPoly with 1-ReLU fails to prove the property whereas 2-ReLU adds extra constraints (in green) that help in verifying the property.

**Third Layer**  Next, the affine assignments $x_7 := x_5 + 2x_6$ and $x_8 := x_6 + 1.5$ are handled. DeepPoly adds the constraints:

$$x_7 \geq x_5 + 2 \cdot x_6, \qquad x_7 \leq x_5 + 2 \cdot x_6, \qquad x_8 \geq x_6 + 1.5, \qquad x_8 \leq x_6 + 1.5, \tag{3}$$

To compute the upper and lower bounds for $x_7$ and $x_8$, DeepPoly substitutes the polyhedral constraints for $x_5$ and $x_6$ from (1) and (2) in (3). It again substitutes for the constraints for $x_5$ and $x_6$ in terms of $x_3$ and $x_4$ and iterates until it reaches the input layer where it substitutes the concrete bounds for $x_1$ and $x_2$. Doing so yields $l_7 = 0, u_7 = 5$ and $l_8 = 1.5, u_8 = 3.5$.

**Refinement with 1-ReLU fails**  Because DeepPoly discards one of the lower bounds from the triangle relaxations for the ReLU assignments in the previous layer, it is possible to refine lower and upper bounds for $x_7$ and $x_8$ by encoding the network upto the final affine transformation using the relatively tighter ReLU relaxations based on the triangle formulation and then computing bounds for $x_7$ and $x_8$ with respect to this formulation. However, this does not improve bounds and still yields $l_7 = 0, u_7 = 5, l_8 = 1.5, u_8 = 3.5$.

As the lower bounds for both $x_7$ and $x_8$ are non-negative, the DeepPoly ReLU approximation simply propagates $x_7$ and $x_8$ to the output layer. The final output is thus:

$$x_9 \geq x_7, \qquad x_9 \leq x_7, \qquad l_9 = 0, \qquad u_9 = 5,$$
$$x_{10} \geq x_8, \qquad x_{10} \leq x_8, \qquad l_{10} = 1.5, \qquad u_{10} = 3.5.$$

Because the upper bound is $u_9 = 5$, the verifier fails to prove the property that $x_9 \leq 4$.

**Refinement with 2-ReLU verifies the property**  Now we consider refinement with our 2-ReLU relaxation which considers the two ReLU assignments $x_5 := ReLU(x_3)$ and $x_6 := ReLU(x_4)$ *jointly*. Besides the box constraints for $x_3$ and $x_4$, it also considers the constraints $x_3 + x_4 \leq 2, x_3 - x_4 \leq -2, -x_3 - x_4 \leq 2, x_4 - x_3 \leq 2$ for computing the output of ReLU. The ReLU output contains the extra constraint $2 \cdot x_5 + 2 \cdot x_6 - x_3 - x_4 \leq 2$ that 1-ReLU cannot capture. We again encode the network upto the final affine transformation with the tighter ReLU relaxations obtained using 2-ReLU and refine the bounds for $x_7, x_8$. Now, we obtain better upper bounds as $u_7 = 4$. The better bound for $u_7$ is then propagated to $u_9$ and is sufficient for proving the desired property.

We remark that while in this work we instantiate the k-ReLU concept with the DeepPoly relaxation, the idea can be applied to other relaxation [11, 7–10, 12, 14, 15, 17, 18].

# 3    k-ReLU relaxation framework

In this section we formally describe our k-ReLU framework for generating optimal convex relaxations in the input-output space for $k$ ReLU operations jointly. In the next section, we discuss the instantiation of our framework with existing verifiers which enables more precise results.

We consider a ReLU based feedforward, convolutional or residual neural network with $h$ neurons from a set $\mathcal{H}$ (that is $h = |H|$) and a bounded input region $\mathcal{I} \subseteq \mathcal{R}^m$ where $m < h$ is the number of neural network inputs. In our exposition, we treat the affine transformation and the ReLUs as separate layers. We consider a convex approximation method $M$ that processes network layers in sequence from the input to the output layer passing the output of predecessor layers as input to the successor layers. Let $\mathcal{S} \subseteq \mathbb{R}^h$ be a convex set computed via $M$ approximating the set of values that neurons upto layer $l$-1 can take with respect to $\mathcal{I}$ and $\mathcal{B} \supseteq \mathcal{S}$ be the smallest bounding box around $\mathcal{S}$. We use $\text{Conv}(\mathcal{S}_1, \mathcal{S}_2)$ and $\mathcal{S}_1 \cap \mathcal{S}_2$ to denote the convex hull and the intersection of convex sets $\mathcal{S}_1$ and $\mathcal{S}_2$, respectively.

Let $\mathcal{X}, \mathcal{Y} \subseteq \mathcal{H}$ be respectively the set of input and output neurons in the $l$-th layer consisting of $n$ ReLU assignments of the form $y_i\text{:=}\text{ReLU}(x_i)$ where $x_i \in \mathcal{X}$ and $y_i \in \mathcal{Y}$. In the general case, each input neuron $x_i$ takes on both positive and negative values in $\mathcal{S}$. We define the polyhedra induced by the two branches of each ReLU assignment $y_i\text{:=}\text{ReLU}(x_i)$ as $\mathcal{C}_i^+ = \{x_i \geq 0, y_i = x_i\} \subseteq \mathbb{R}^h$ and $\mathcal{C}_i^- = \{x_i \leq 0, y_i = 0\} \subseteq \mathbb{R}^h$. Let $\mathcal{Q}_J = \{\bigcap_{i \in J} C_i^{s(i)} \mid s \in J \to \{-,+\}\}$ (where $J \subseteq [n]$) be the set of polyhedra $Q \subseteq \mathbb{R}^h$ constructed by the intersection of polyhedra $\mathcal{C}_i \subseteq \mathbb{R}^h$ for neurons $x_i, y_i$ indexed by the set $J$ such that each $\mathcal{C}_i \in \{\mathcal{C}_i^+, \mathcal{C}_i^-\}$. We next formulate the best convex relaxation of the output after $n$ ReLU assignments.

## 3.1    Best convex relaxation

The best convex relaxation after the $n$ ReLU assignments is given by

$$\mathcal{S}_{\text{best}} = \text{Conv}_{Q \in \mathcal{Q}_{[n]}}(Q \cap \mathcal{S}). \tag{4}$$

$\mathcal{S}_{\text{best}}$ considers all $n$ assignments jointly. Computing it is practically infeasible as it involves computing $2^n$ convex hulls each of which has exponential cost in the number of neurons $h$ [24].

## 3.2    1-ReLU

We now describe the prior convex relaxation [3] through triangles (here called 1-ReLU) that handles the $n$ ReLU assignments *separately*. Here, the input to the $i$-th assignment $y_i\text{:=}\text{ReLU}(x_i)$ is the polyhedron $P_{\text{1-ReLU}} \supseteq \mathcal{S}$ where for each $x_i \in \mathcal{X}$, $P_{\text{1-ReLU},i}$ contains only an interval constraint $[l_i, u_i]$ that bounds $x_i$, that is, $l_i \leq x_i \leq u_i$. Here, the interval bounds are simply obtained from the bounding box $\mathcal{B}$ of $\mathcal{S}$. The output of this method after $n$ assignments is

$$\mathcal{S}_{\text{1-ReLU}} = \mathcal{S} \cap \bigcap_{i=1}^{n} \text{Conv}(P_{\text{1-ReLU},i} \cap \mathcal{C}_i^+, P_{\text{1-ReLU},i} \cap \mathcal{C}_i^-). \tag{5}$$

The projection of $\text{Conv}(P_{\text{1-ReLU},i} \cap \mathcal{C}_i^+, P_{\text{1-ReLU},i} \cap \mathcal{C}_i^-)$ onto the $x_i y_i$-plane is a triangle minimizing the area and is the optimal convex relaxation in this plane. However, because the input polyhedron $P_{\text{1-ReLU}}$ is a hyperrectangle (when projected to $\mathcal{X}$), it does not capture relational constraints between different $x_i$'s in $\mathcal{X}$ (meaning it typically has to substantially over-approximate the set $\mathcal{S}$). Thus, as expected, the computed result $\mathcal{S}_{\text{1-ReLU}}$ of the 1-ReLU method will incur significant imprecision when compared with the $S_{\text{best}}$ result.

## 3.3    k-ReLU relaxations

We now describe our k-ReLU framework for computing a convex relaxation of the output of $n$ ReLUs in one layer by considering groups of $k$ ReLUs jointly with $k > 1$. For simplicity, we assume that $n > k$ and $k$ divides $n$. Let $\mathcal{J}$ be a partition of the set of indices $[n]$ such that each block $\mathcal{J}_i \in \mathcal{J}$ contains exactly $k$ indices. Let $P_{\text{k-ReLU},i} \subseteq \mathbb{R}^h$ be a polyhedron containing interval and relational constraints over the neurons from $\mathcal{X}$ indexed by $\mathcal{J}_i$. In our framework, $P_{\text{k-ReLU},i}$ is derived via $\mathcal{B}$ and $\mathcal{S}$ and satisfies $\mathcal{S} \subseteq P_{\text{k-ReLU},i}$.

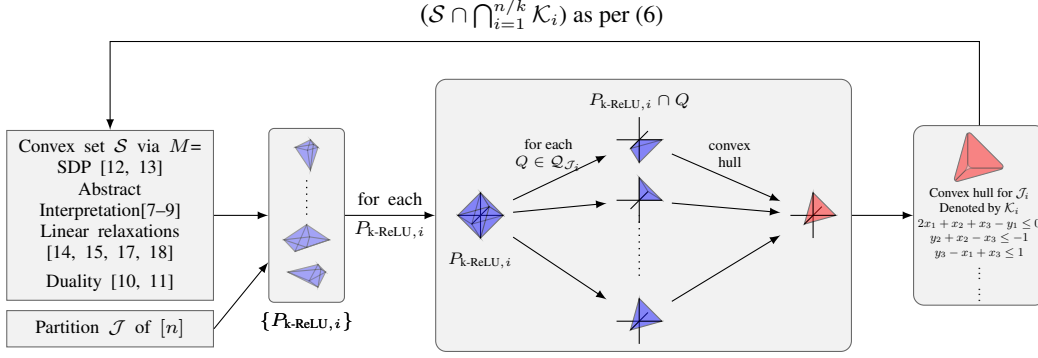

Figure 3: Steps to instantiating the k-ReLU framework.

Our k-ReLU framework produces the following convex relaxation of the output:

$$\mathcal{S}_{\text{k-ReLU}} = \mathcal{S} \cap \bigcap_{i=1}^{n/k} \text{Conv}_{Q \in \mathcal{Q}_{\mathcal{J}_i}} (P_{\text{k-ReLU},i} \cap Q). \tag{6}$$

The result of (6) is the optimal convex relaxation for the output of $n$ ReLUs for the given choice of $\mathcal{S}, k, \mathcal{J}$, and $P_{\text{k-ReLU},i}$.

**Theorem 3.1.** *For $k > 1$ and a partition $\mathcal{J}$ of indices, if there exists a $\mathcal{J}_i$ for which $P_{\text{k-ReLU},i} \subsetneq \bigcap_{u \in \mathcal{J}_i} P_{\text{l-ReLU},u}$ holds, then $\mathcal{S}_{\text{k-ReLU}} \subsetneq \mathcal{S}_{\text{l-ReLU}}$.*

The proof of Theorem 3.1 is given in appendix. Note that $P_{\text{l-ReLU}}$ only contains interval constraints whereas $P_{\text{k-ReLU}}$ contains both, the same interval constraints and extra relational constraints. Thus, any convex relaxation obtained using k-ReLU is typically strictly more precise than a 1-ReLU one.

**Precise and scalable relaxations for large** $k$ For each $\mathcal{J}_i$, the optimal convex relaxation $\mathcal{K}_i = \text{Conv}_{Q \in \mathcal{Q}_{\mathcal{J}_i}} (P_{\text{k-ReLU},i} \cap Q)$ from (6) requires computing the convex hull of $2^k$ convex sets each of which has a worst-case exponential cost in terms of $k$. Thus, computing $\mathcal{K}_i$ via (6) can become computationally expensive for large values of $k$. We propose an efficient relaxation $\mathcal{K}'_i$ for each block $\mathcal{J}_i \in \mathcal{J}$ (where $|\mathcal{J}_i| = k$ as described earlier) based on computing relaxations for all subsets of $\mathcal{J}_i$ that are of size $2 \le l < k$. Let $\mathcal{R}_i = \{\{j_1, \ldots, j_l\} \mid j_1, \ldots, j_l \in \mathcal{J}_i\}$ be the set containing all subsets of $\mathcal{J}_i$ containing $l$ indices. For each $R \in \mathcal{R}_i$, let $P'_{\text{l-ReLU},R} \subseteq \mathbb{R}^h$ be a polyhedron containing interval and relational constraints between the neurons from $\mathcal{X}$ indexed by $R$ with $\mathcal{S} \subseteq P'_{\text{l-ReLU},R}$.

The relaxation $\mathcal{K}'_i$ is computed by applying $l$-ReLU $\binom{k}{l}$ times as:

$$\mathcal{K}'_i = \bigcap_{R \in \mathcal{R}_i} \text{Conv}_{Q \in \mathcal{Q}_R} (P'_{\text{l-ReLU},R} \cap Q). \tag{7}$$

The layerwise convex relaxation $\mathcal{S}'_{\text{k-ReLU}} = \mathcal{S} \cap \bigcap_{i=1}^{n/k} \mathcal{K}'_i$ via (7) is tighter than computing relaxation $\mathcal{S}_{\text{l-ReLU}}$ via (6) with a partition $\mathcal{J}'$ where for each block $\mathcal{J}'_i \in \mathcal{J}'$ there exists $\mathcal{R}_j$ corresponding to a block of $\mathcal{J}$ such that $\mathcal{J}'_i \in \mathcal{R}_j$ and $P'_{\text{l-ReLU},\mathcal{J}'_i} \subseteq P_{\text{l-ReLU},\mathcal{J}'_i}$ where $P_{\text{l-ReLU},\mathcal{J}'_i}$ is the polyhedron in (6) for computing $\mathcal{S}_{\text{l-ReLU}}$. In our instantiations, we ensure that this condition holds for gaining precision.

## 4 Instantiating the k-ReLU framework

Our k-ReLU framework from Section 3 can be instantiated to produce different relaxations depending on the parameters $\mathcal{S}, k, \mathcal{J}$, and $P_{\text{k-ReLU},i}$. Fig. 3 shows the steps to instantiating our framework. The inputs to the framework is the convex set $\mathcal{S}$ and the partition $\mathcal{J}$ based on $k$. These inputs are first used to produce a set containing $n/k$ polyhedra $\{P_{\text{k-ReLU},i}\}$. Each polyhedron $P_{\text{k-ReLU},i}$ is then intersected with polyhedra from the set $\mathcal{Q}_{\mathcal{J}_i}$ producing $2^k$ polyhedra which are then combined via the convex hull (each called $\mathcal{K}_i$). The $\mathcal{K}_i$'s are then combined with $\mathcal{S}$ to produce the final relaxation that captures the values which neurons can take after the ReLU assignments. This relaxation is tighter than that produced by applying $M$ directly on the ReLU layer enabling precision gains.

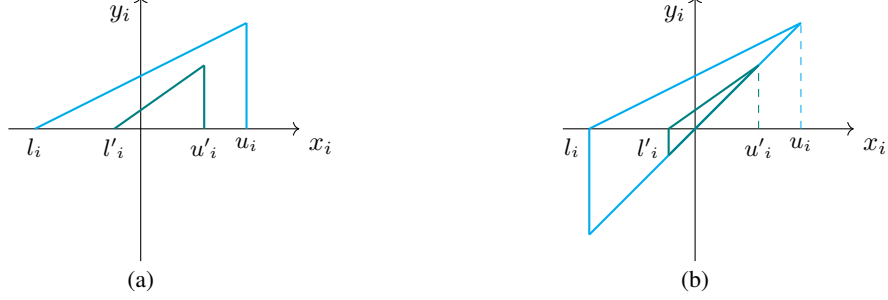

<div style="text-align:center">(a)           (b)</div>

Figure 4: DeepPoly relaxations for $y_i$:=ReLU($x_i$) using the original bounds $l_i, u_i$ (in blue) and the refined bounds $l'_i, u'_i$ (in green) for $x_i$. The refined relaxations have smaller area in the $x_i y_i$-plane.

## 4.1 Computing key parameters

We next describe the choice of the key parameters $\mathcal{S}, k, \mathcal{J}, P_{\text{k-ReLU},i}$ in our framework.

**Input convex set** Examples of convex approximation method $M$ for computing $\mathcal{S}$ include [11, 7–10, 12, 14, 15, 17, 18]. In this paper, we use the DeepPoly [9] relaxation for computing $\mathcal{S}$ which is a state-of-the-art precise and scalable verifier for neural networks.

$k$ **and partition** $\mathcal{J}$ We use (6) to compute the output relaxation when $k \in \{2,3\}$. For larger $k$, we compute the output based on (7). To maximize precision gain, we group those indices $i$ together into a block where the triangle relaxation for $y_i$:=ReLU($x_i$) has the larger area in the $x_i y_i$-plane.

**Computing $P_{\text{krelu},i}$** We note that for a fixed block $\mathcal{J}_i$, several polyhedron $P_{\text{k-ReLU},i}$ are possible that produce convex relaxations with varying degree of precision. Ideally, one would like $P_{\text{k-ReLU},i}$ to be the projection of $\mathcal{S}$ onto the variables in the set $\mathcal{X}$ indexed by the block $\mathcal{J}_i$. However, computing this projection exactly is expensive and therefore we compute a relaxation of it.

We use the method $M$ to compute $P_{\text{k-ReLU},i}$ by computing the upper bounds for linear relational expressions of the form $\sum_{u=1}^{k} a_u \cdot x_u$ with respect to $\mathcal{S}$. In our experiments, we found that setting $a_u \in \{-1, 0, 1\}$ yields maximum precision (except the case where all possible $a_u$ are zero). Thus $P_{\text{k-ReLU},i} \supseteq \mathcal{S}$ contains $3^k - 1$ constraints which include the interval constraints for all $x_u$.

## 4.2 Verification with k-ReLU framework

Let $\psi \subseteq \mathbb{R}^h$ be a convex set defining a safe region for the outputs with respect to the input region $\mathcal{I}$ and $\mathcal{S}_O \subseteq \mathbb{R}^h$ be the output convex relaxation obtained after processing affine layers with the convex approximation method $M$ and ReLU layers with our k-ReLU framework. $\psi$ holds if $\psi \subseteq \mathcal{S}_O$.

## 4.3 Instantiation with DeepPoly

We now show to instantiate the k-ReLU framework with DeepPoly [9]. DeepPoly is a type of a restricted Polyhedra abstraction which balances scalability and precision of the analysis. It associates four constraints per neuron $h_i \in \mathcal{H}$: (a) a lower polyhedral constraint of the form $a_i^{\leq} \leq h_i$, (b) an upper polyhedral constraint $h_i \leq a_i^{\geq}$, (c) a lower bound constraint $l_i \leq h_i$, and (d) an upper bound constraint $h_i \leq u_i$. The polyhedral expressions $a_i^{\leq}, a_i^{\geq}$ are of the form $\sum_j a_j \cdot h_j + c$ where $a_j, c \in \mathbb{R}$ and capture relational information ensuring that DeepPoly is exact for affine transformations. The analysis proceeds layer by layer and thus the polyhedral constraints for a neuron in layer $l$ contain only the neurons upto layer $l$-1. $\mathcal{S}$ here is the set of points satisfying DeepPoly constraints for all neurons. We next discuss how the k-ReLU framework can be used for improving the precision of the ReLU transformer for DeepPoly and also that of the overall verification procedure.

**Improving the precision of DeepPoly ReLU relaxation** DeepPoly loses precision for ReLU assignments $y_i$:=ReLU($x_i$) where $x_i$ can take both positive and negative values. It computes convex relaxations shown in Fig. 4 (a) and (b). It keeps the one with smaller area in the $x_i y_i$-plane.We note that both of these relaxations depend only on the interval bounds $l_i, u_i$ for $x_i$. DeepPoly uses backsubstitution (see [9] for details) for obtaining precise bounds $l_i, u_i$. We note that DeepPoly

<div style="text-align:center">7</div>

Table 2: Neural network architectures and parameters used in our experiments.

| Dataset | Model | Type | #Neurons | #Layers | Defense | Refine ReLU | $k$ |
|---------|-------|------|----------|---------|---------|-------------|-----|
| MNIST | $6 \times 100$ | fully connected | 610 | 6 | None | ✓ | 3 |
| | $9 \times 100$ | fully connected | 910 | 9 | None | ✓ | 2 |
| | $6 \times 200$ | fully connected | 1 210 | 6 | None | ✓ | 2 |
| | $9 \times 200$ | fully connected | 1 810 | 9 | None | ✓ | 2 |
| | ConvSmall | convolutional | 3 604 | 3 | None | ✗ | Adaptive |
| | ConvBig | convolutional | 34 688 | 6 | DiffAI [29] | ✗ | 5 |
| CIFAR10 | ConvSmall | convolutional | 4 852 | 3 | PGD [30] | ✗ | Adaptive |
| | ConvBig | convolutional | 62,464 | 6 | PGD [30] | ✗ | 5 |
| | ResNet | Residual | 107,496 | 13 | Wong [11] | ✗ | Adaptive |

ReLU relaxations are weaker than the 1-ReLU relaxation (as they contain one constraint less than the triangle). This precision loss accumulates as the analysis proceeds deeper in the network. We now show how the k-ReLU framework can recover precision for DeepPoly.

We compute refined bounds $l'_i, u'_i$ for those neurons $x_i$ that are inputs to a ReLU and can take positive values. We maximize and minimize $x_i$ with respect to the convex relaxation produced by replacing the DeepPoly ReLU relaxation (Fig. 4 (a) and (b)) with our k-ReLU relaxations based on (6). Since the constraints from both *DeepPoly* and k-ReLU are linear, we use a LP solver for maximizing and minimizing. $l'_i$ and $u'_i$ facilitate the two DeepPoly ReLU relaxations shown in green in Fig. 4 (a) and (b). These relaxations are tighter than the original ones and improve the precision of DeepPoly.

**k-ReLU for improving robustness certification** When DeepPoly alone cannot prove the target property, we instead check if $\psi$ holds with the tighter ReLU relaxations from k-ReLU via LP solver.

## 5 Evaluation

We instantiated our k-ReLU framework with DeepPoly in the form of a verifier called *kPoly*. *kPoly* is written in Python and uses cdd [25, 26] for computing convex hulls, and Gurobi [27] for refining DeepPoly ReLU relaxations and proving that $\psi$ holds with k-ReLU relaxations. We made *kPoly* publicly available as part of the ERAN [28] framework for neural network verification. We evaluated *kPoly* for the task of robustness certification of challenging deep neural networks. We compare *kPoly* against two state-of-the-art verifiers: DeepPoly [9] and *RefineZono* [19]. DeepPoly has the same precision as [15, 16] whereas *RefineZono* refines the results of *DeepZ* [8] and is more precise than [8, 11, 14]. Both, DeepPoly and *RefineZono* are more scalable than [5, 18, 12, 4, 13, 10], however we show that *kPoly* is more precise than DeepPoly and *RefineZono* while also scaling to large networks. We next describe the neural networks, benchmarks and parameters used in our experiments.

**Neural networks** We used 9 MNIST [31] and CIFAR10 [32] fully connected (FNNs), convolutional (CNNs), and residual networks with ReLU activations shown in Table 2. The first 8 networks in Table 2 are available at https://github.com/eth-sri/eran; the residual network is taken from https://github.com/locuslab/convex_adversarial. Five of the networks do not use adversarial training while the rest use different variants of it. The MNIST ConvBig network is trained with DiffAI [29], the two CIFAR10 convolutional networks are trained with PGD [30] and the residual network is trained via [11]. The largest network in our experiments contains $> 100$K neurons and has 13 layers.

**Robustness property** We consider the $L_\infty$-norm [33] based adversarial region around a correctly classified image from the test set parameterized by the radius $\epsilon \in \mathbb{R}$. Our goal is to certify that the network classifies all images in the adversarial region correctly.

**Machines** The runtimes of all experiments for the MNIST FNNs were measured on a 3.3 GHz 10 Core Intel i9-7900X Skylake CPU with a main memory of 64 GB whereas the experiments for the rest were run on a 2.6 GHz 14 core Intel Xeon CPU E5-2690 with 512 GB of main memory.

Table 3: Number of verified adversarial regions and runtime of *kPoly* vs. DeepPoly and RefineZono.

| Dataset | Model | #correct | $\epsilon$ | DeepPoly[9] | | *RefineZono* [19] | | *kPoly* | |
|---------|-------|----------|------------|-------------|---------|-------------------|---------|---------|---------|
| | | | | verified(#) | time(s) | verified(#) | time(s) | verified(#) | time(s) |
| MNIST | $6 \times 100$ | 960 | 0.026 | 160 | 0.3 | 312 | 310 | 441 | 307 |
| | $9 \times 100$ | 947 | 0.026 | 182 | 0.4 | 304 | 411 | 369 | 171 |
| | $6 \times 200$ | 972 | 0.015 | 292 | 0.5 | 341 | 570 | 574 | 187 |
| | $9 \times 200$ | 950 | 0.015 | 259 | 0.9 | 316 | 860 | 506 | 464 |
| | ConvSmall | 980 | 0.12 | 158 | 3 | 179 | 707 | 347 | 477 |
| | ConvBig | 929 | 0.3 | 711 | 21 | 648 | 285 | 736 | 40 |
| CIFAR10 | ConvSmall | 630 | 2/255 | 359 | 4 | 347 | 716 | 399 | 86 |
| | ConvBig | 631 | 2/255 | 421 | 43 | 305 | 592 | 459 | 346 |
| | ResNet | 290 | 8/255 | 243 | 12 | 243 | 27 | 245 | 91 |

**Benchmarks** For each MNIST and CIFAR10 network, we selected the first 1000 images from the respective test set and filtered out incorrectly classified images. The number of correctly classified images by each network are shown in Table 3. We chose challenging $\epsilon$ values for defining the adversarial region for each network. We note that our benchmarks (e.g., the $9 \times 200$ network with $\epsilon = 0.015$) are quite challenging to handle for state-of-the-art verifiers (as we will see below).

**k-ReLU parameters for the experiments** We refine both the DeepPoly ReLU relaxation and the verification results for the MNIST FNNs whereas only the verification results are refined for the rest. All neurons that can take positive values after the affine transformation are selected for refinement. As an optimization, we use the MILP ReLU encoding from [5] when refining the ReLU relaxation for the second ReLU layer. The last column of Table 2 shows the value of $k$ for all networks. For the MNIST and CIFAR10 ConvBig networks, we encode the first 3 ReLU layers with 1-ReLU while the remaining are encoded with 5-ReLU. We use $l = 3$ in (7) for encoding 5-ReLU. For the remaining 3 networks, we encode the first ReLU layer with 1-ReLU while the remaining layers are encoded adaptively. Here, we choose a value of $k$ for which the total number of calls to 3-ReLU is $\leq 500$. We next discuss our experimental results shown in Table 3.

*kPoly* **vs DeepPoly and** *RefineZono* Table 3 compares the precision in number of adversarial regions verified and the average runtime per image in seconds for *kPoly*, DeepPoly and *RefineZono*. We refine the verification results with *RefineZono* and *kPoly* only when *DeepZ* and DeepPoly fails to verify. *kPoly* is more precise than both DeepPoly and *RefineZono* on all networks. *RefineZono* is more precise than DeepPoly on the networks trained without adversarial training. On the $9 \times 200$ and MNIST ConvSmall networks, *kPoly* verifies 506 and 347 regions respectively whereas *RefineZono* verifies 316 and 179 regions respectively. The precision gain is less on networks with adversarial training and *kPoly* verifies 25, 40, 38 and 2 regions more than DeepPoly on the last 4 networks in Table 3. *kPoly* is faster than *RefineZono* on all networks and has an average runtime of $< 8$ minutes. The largest runtimes are on the MNIST $9 \times 200$ and ConvSmall networks. These are quite small compared to the CIFAR10 ResNet network where *kPoly* has an average runtime of only 91 seconds.

**1-ReLU vs k-ReLU** We consider the first 100 regions for the MNIST ConvSmall network and compare the number of regions verified by *kPoly* when run with k-ReLU and 1-ReLU. We note that *kPoly* run with 1-ReLU is equivalent to [17]. *kPoly* with 1-ReLU verifies 20 regions whereas with k-ReLU it verifies 35. *kPoly* with 1-ReLU has an average runtime of 9 seconds.

**Effect of heuristic for** $\mathcal{J}$ We ran *kPoly* based on k-ReLU with random partitioning $\mathcal{J}_r$ using the same setup as for 1-ReLU. We observed that *kPoly* produced worse bounds and verified 34 regions.

# 6   Conclusion

We presented k-ReLU, a novel parametric framework which produces more precise results than the single neuron triangle convex relaxation. The key idea of k-ReLU is to consider multiple ReLUs *jointly*. We showed k-ReLU leads to significantly improved precision, enabling us to prove properties beyond the reach of prior work, while preserving scalability.

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

# A  Appendix

## A.1  Proof of Theorem 3.1

*Proof.* Since $P_{\text{k-ReLU},i} \subsetneq \bigcap_{u \in \mathcal{J}_i} P_{\text{1-ReLU},u}$ for $\mathcal{J}_i$, by monotonicity of intersection and convex hull,

$$\text{Conv}_{Q \in \mathcal{Q}_{\mathcal{J}_i}}(P_{\text{k-ReLU},i} \cap Q) \subsetneq \text{Conv}_{Q \in \mathcal{Q}_{\mathcal{J}_i}}((\bigcap_{u \in \mathcal{J}_i} P_{\text{1-ReLU},u}) \cap Q) \tag{8}$$

For any $Q \in \mathcal{Q}_{\mathcal{J}_i}$, we have that either $Q \subseteq \mathcal{C}_u^+$ or $Q \subseteq \mathcal{C}_u^-$ for $u \in \mathcal{J}_i$. Thus, we can replace all $Q$ on the right hand side of (8) with either $\mathcal{C}_u^+$ or $\mathcal{C}_u^-$ such that for all $u \in \mathcal{J}_i$ both $\mathcal{C}_u^+$ and $\mathcal{C}_u^-$ are used at least in one substitution and obtain by monotonicity,

$$\subseteq \text{Conv}_{u \in \mathcal{J}_i}((\bigcap_{u \in \mathcal{J}_i} P_{\text{1-ReLU},u}) \cap \mathcal{C}_u^+, (\bigcap_{u \in \mathcal{J}_i} P_{\text{1-ReLU},u}) \cap \mathcal{C}_u^-)$$

$$\subseteq \text{Conv}_{u \in \mathcal{J}_i}(P_{\text{1-ReLU},u} \cap \mathcal{C}_u^+, P_{\text{1-ReLU},u} \cap \mathcal{C}_u^-) \qquad (\bigcap_{u \in \mathcal{J}_i} P_{\text{1-ReLU},u} \subseteq P_{\text{1-ReLU},u}).$$

For remaining $i$, similarly $\text{Conv}_{Q \in \mathcal{Q}_i}(P_{\text{k-ReLU},i} \cap Q) \subseteq \text{Conv}_{u \in \mathcal{J}_i}(P_{\text{1-ReLU},u} \cap \mathcal{C}_u^+, P_{\text{1-ReLU},u} \cap \mathcal{C}_u^-)$ holds. Since $\subsetneq$ relation holds for at least one $i$ and $\subseteq$ holds for others, $\mathcal{S}_{\text{k-ReLU}} \subsetneq \mathcal{S}_{\text{1-ReLU}}$ holds. $\square$

