[Supplementary Material · neurips_2019_appendix.pdf]


# A  Appendix

## A.1  Proof of Theorem 3.1

*Proof.* Since $P_{\text{k-ReLU},i}$ contains $\mathcal{P}_{R,i} \neq \emptyset$ and both $P_{\text{k-ReLU},i}$, $\prod_{i=1}^{k} P_{\text{1-ReLU},\mathcal{I}_i}$ contain the same

interval constraints, we have that $P_{\text{k-ReLU},i} \subsetneq \prod_{i=1}^{k} P_{\text{1-ReLU},\mathcal{I}_i}$. We have by monotonicity of the

intersection and the convex hull,

$$( \bigsqcup_{S_i \in \mathcal{S}_i} P_{\text{k-ReLU},i} \sqcap S_i) \subsetneq ( \bigsqcup_{S_i \in \mathcal{S}_i} ( \prod_{u \in \mathcal{I}_i} P_{\text{1-ReLU},u}) \sqcap S_i) \tag{9}$$

For any $S_i \in \mathcal{S}_i$, we have that either $S_i \subseteq \mathcal{C}_u^+$ or $S_i \subseteq \mathcal{C}_u^-$ for any $u \in \mathcal{I}_i$. Thus, we can replace all

$S_i$ on the right hand side of (9) with either $\mathcal{C}_u^+$ or $\mathcal{C}_u^-$ such that for all $u \in \mathcal{I}_i$ both $\mathcal{C}_u^+$ and $\mathcal{C}_u^-$ are

used in at least one substitution and obtain by monotonicity,

$$\subseteq ( \bigsqcup_{j=1}^{k} (( \prod_{u \in \mathcal{I}_i} P_{\text{1-ReLU},u}) \sqcap \mathcal{C}_u^+) \sqcup (( \prod_{u \in \mathcal{I}_i} P_{\text{1-ReLU},u}) \sqcap \mathcal{C}_u^-))$$

$$\subseteq ( \bigsqcup_{u \in \mathcal{I}_i} (P_{\text{1-ReLU},u} \sqcap \mathcal{C}_u^+) \sqcup (P_{\text{1-ReLU},u} \sqcap \mathcal{C}_u^-)).$$

For other $i$'s, it can be shown similarly that $( \bigsqcup_{S_i \in \mathcal{S}_i} P_{\text{k-ReLU},i} \sqcap S_i) \subseteq ( \bigsqcup_{u \in \mathcal{I}_i} (P_{\text{1-ReLU},u} \sqcap \mathcal{C}_u^+) \sqcup$

$(P_{\text{1-ReLU},u} \sqcap \mathcal{C}_u^-))$ holds. Since $\subsetneq$ relation holds for at least one $i$ and $\subseteq$ holds for other $i$'s, $P_{\text{k-ReLU}} \subsetneq$

$P_{\text{1-ReLU}}$ holds by the order preserving property of the intersection. □

## A.2  1-ReLU vs 2-ReLU vs 3-ReLU on the $9 \times 200$ network

We graphically show the precision and the average runtime of 1-ReLU, 2-ReLU, and 3-ReLU with no

MILP and with one layer of MILP on the $9 \times 200$ MNIST network with $\epsilon = 0.015$ using the results
from the evaluation section in Fig. 4.

Figure 4: Percentage of images verified and the average runtime (in seconds) of 1-ReLU, 2-ReLU, and 3-ReLU on the $9 \times 200$ MNIST network with $\epsilon = 0.015$.