[Reviews · NeurIPS 2019]

Reviewer 1



Originality: The authors propose a novel relaxation (to the best of my knowledge) for networks with ReLU activations that tighten previously proposed relaxations that ignore the correlations between neurons in the network. The theoretical results are also novel (although unsurprising). However, it would be useful for the authors to better clarify the computational requirements and tightness of k-ReLU relative to DeepPoly and other similar relaxations and bound propagation methods like [13] and https://arxiv.org/abs/1805.12514, especially the approximate version (equation (7)) in the paper. Quality: The theoretical results are accurate (albeit unsurprising) in my opinion. The experimental section is missing several important details in my opinion: 1) The authors say that experiments are performed on both MNIST and CIFAR-10, but the tables 2/3 only report numbers on MNIST. What about the results on CIFAR10? It would be important to see these to understand how well the method scales to a larger dataset. 2) The sizes of neural networks studied in the paper are still tiny relative to networks that have been considered by previous work on neural network verification: for example, the works from https://arxiv.org/pdf/1805.12514.pdf (which has 107496 hidden units, almost 3x the size of the largest network studied in the paper). 3) The authors only report the time limit used for the LP and MILP solvers in their framework. However, these are just subroutines in the overall verification procedure and it would be important to see the total verification time and how it scales with network size. This is particularly important since the method requries solving an LP for each neuron to be tightened - a cost that can grow superlinarly with network size rendering the method ineffective. 4) The authors do not compare against many other classes of methods that have been shown to be effective at scalable neural network verification (in particular, I think comparisons against [13] and https://arxiv.org/abs/1805.12514 would be instructive). 5) It would be instructive to have precise timing comparisons between methods and also study how these comparisons scale with increasing network width/depth etc. Clarity: I think the paper can be reorganized for better readability. Section 2 does not provide much insight in my mind - it would be preferable to explain the relaxation tightening geometrically rather than through a numerical example. Further, rather than making extensive the comparison to DeepPoly, it would be good if the authors addressed the broader literature in the field and qunatified the precise gains from kPoly over methods used in the literature. Significance: I believe that the results are potentially significant, offering a way to interpolate between scalable but conservative verification methods and exact but computationally intractable verification methods. However, I do think that the experiments and connections/distinctions from prior work (particularly in terms of computational complexity) need to be fleshed out further to make the results in the paper stronger.

Reviewer 2



My summary of the paper: Most relaxation-based methods for certifying the robustness of neural networks use the "triangle approximation," which adds a constraint between each pre-ReLU activation and the corresponding post-ReLU activation. Unfortunately, by only treating each unit independently, the triangle approximation loses a lot of precision, i.e. the relaxation is very loose. Consider a ReLU layer with n units. The tightest possible convex relaxation is the convex hull, which is described by 2^n sets of linear inequalities, one for each possible "activation pattern" (an activation pattern is a choice of - or + for each ReLU). Enumerating exponentially many inequalities is computationally intractable, so the authors instead propose the following "K-ReLU" relaxation: arbitrarily partition the units in the layer into mutually exclusive sets of size K, and add the linear inequalities corresponding to the convex hull of each K-sized set. Apparently, this too is computationally intractable in practice for K > 2, so when K > 2, the authors propose to use a further relaxation which (I think) only adds linear inequalities that involve at most two units. The authors employ this K-ReLU relaxation as a way to tighten the bounds for DeepPoly, a neural network certification method; to be honest, though, I didn't understand all the details involved (see below). Remark: - I do not understand where exactly DeepPoly enters into the proposed method. From what I gather, you compute the lower and upper interval bounds l'_x and u'_x using a linear programming solver. Now, you _could_ use that LP solver to also compute the 'intercepts' for the octagonal constraints. But you apparently use DeepPoly for that --- why? It's unclear to me why DeepPoly is necessary at all. (Is it because of speed?) Moreover, it's also unclear to me where the DeepPoly ordering restriction (that each unit's constraint only depend on the preceding units) enters into the proposed algorithm. There was some confusing notation in this paper: - on line 136, 148, and elsewhere, the authors speak of the "union" of constraints when they actually mean the intersection. (I understand why "union" arguably applies too, I just think it's confusing, given that union and intersection are opposites). Also, I believe that in Equation 8, this is explicitly wrong -- you need to use \cap instead of \cup. - in line 142, the authors define "boxy \cap" (I don't know the actual latex command) to mean the intersection of two polyhedra, and "boxy \cup" to mean the convex hull of two polyhedra. This is confusing for three reasons: (1) boxy \cup looks very similar to \cup, yet you use it to mean something very different (convex hull vs. union); (2) in your definitions, there's a weird asymmetry between boxy \cap and boxy \cup; (3) in your definitions, boxy \cap is the same thing as \cap. Here is my suggestion: just use \cap and \cup to mean intersection and union, and use conv() to mean the convex hull of a set. - on line 151, it's not "2^n convex hulls", it's one convex hull that requires 2^n inequalities - in Theorem 3.1, I have no idea what that symbol means -- you should define it in the text. - on line 147, each S is actually a subset of R^{h x h} and not (as is stated) R^h, right? Advantages of the proposed method: - can be used to obtain superior to robustness certificates to DeepPoly and RefineZono. I don't understand how these numbers stack up to those from other methods, e.g. https://arxiv.org/abs/1805.12514 or https://arxiv.org/abs/1810.12715, but comparisons to other methods might not be necessary, since the K-ReLU idea is generic and could be used with other relaxation-based certification methods. Disadvantages of the proposed method: - requires arbitrary choice for the partitioning of ReLUs in each layer - requires arbitrary choice for P_{R, i} Rubric: - originality: the idea of tightening relaxation-based bounds by considering multiple ReLUs is a natural one, but to my knowledge, it has not been thoroughly explored in the literature yet. Therefore, this work is original. - quality: the paper is of good quality. - clarity: as I've mentioned in this review, the clarity can definitely be improved, but the paper is certainly readable. - significance: I am not quite sure of the significance of this work to the literature. Therefore, I am rating my confidence in this review as low. === response ===== I've read the author rebuttal. The other reviewers agree that the notation is at a minimum confusing, and possibly wrong. I encourage the authors to improve the clarity of the mathematical presentation in the revision.

Reviewer 3



This work introduces an improved framework that allows abstraction based certifiers to compute the optimal convex relaxation over many ReLU operations jointly, as opposed to relaxing each ReLU individually (which had been the focus of much prior work). Overall, the work is clearly written and organized, presents a clear example illustrating the benefit of this improved approach, and has a solid empirical comparison to some prior work that shows reasonable improvement for some tasks and minor improvements for other tasks. While I tend to vote in favor of accepting this work, I do have some suggestions for improving the paper, and would like to see these comments addressed as well as possible. Defining the word “precision” - the word “precision” is used many times throughout the paper, but it is not defined. It may be worth defining to avoid confusion with the definition of precision corresponding to “precision vs. recall.” The word is most confusing in Table 3. The column labeled “precision (%)” does not make sense. My best interpretation of the table is that the “precision (%)” column means the % of images certified to be robust. If so, that should be corrected. Baselines - As far as I’m aware, the most common baselines for certification methods are the MNIST ConvSmall/ConvBig networks and the CIFAR10 ConvSmall network. While I see an improvement over related abstraction-based methods, the results would be more compelling if a comparison can be made to common MILP and LP-based approaches (e.g. [1] for MILP or FastLin/[2] for LP-based approaches. They all have open source code). This is especially important because the kPoly method uses a mixture of LP and MILP solvers, so it would be important to understand the tradeoff when you use a purely LP-based or a purely MILP-based approach. As it is, kPoly already seems relatively slow (and thus, maybe not extremely scalable), so it’s important to see how much benefit it provides over the faster LP approach, or to compare it to a slower exact method like MILP to see how close it is to the exactness of MILP. If DeepPoly is already implementing an LP-based approach, that’s worth clarifying too. - Additionally, robustness certification is most commonly performed with the goal of certifying a relatively robust network, and achieving high provably robust accuracy. Even though the results in Table 3 show an improvement over DeepPoly and RefineZono, the final numbers (45% for eps=0.3 on MNIST, and 23% for eps=0.03 on CIFAR10) are very subpar compared to related works like [1, 2, 3]. If that is the best result that is achievable by this method, I am worried about its usefulness relative to other approaches. Thus, I believe it’s worth trying to apply kPoly to networks where the best known provably robust accuracy is much higher (e.g. the open source models from https://github.com/locuslab/convex_adversarial/tree/master/models_scaled) - Finally, I’d be interested to know if this approach can scale to larger architectures like ResNet, which some other certification techniques do appear capable of handling. Missing Related work - I believe that [2, 3] are important related papers that provide good benchmarks for comparison, and should be cited. Finally, the work of [4] may be quite relevant as well. It discusses exactly the idea of finding MIP formulations that are optimal when considering multiple ReLUs at a time, and it applies to the problem of verifying ReLU neural networks as well. Additional Appendix Information - I would appreciate it if the authors could explain how the expression in section 3.3 leads to the simple linear constraints that 2-ReLU uses in Figure 2. I personally couldn’t follow how to do so. Finally, some minor comments Line 88 - Should be Fig. 2, not Fig 3. Line 107 - Is 1-ReLU equivalent to the LP-based formulation of FastLin/[2]? It’s worth explaining how they are related. Figure 3 - Why are there two figures? What are (a) and (b)? Table 2/Table 3 - I think the final row is missing the dataset (CIFAR10) [1] Tjeng, V., Xiao, K. Y., and Tedrake, R. Evaluating robustness of neural networks with mixed integer programming. In International Conference on Learning Representations, 2019. URL https://openreview.net/forum?id=HyGIdiRqtm. [2] Wong, E. and Kolter, Z. Provable defenses against adversarial examples via the convex outer adversarial polytope. In International Conference on Machine Learning (ICML), pp. 5283–5292, 2018. [3] Gowal, S., Dvijotham, K., Stanforth, R., Bunel, R., Qin, C., Uesato, J., Mann, T., and Kohli, P. On the effectiveness of interval bound propagation for training verifiably robust models. arXiv preprint arXiv:1810.12715, 2018. [4] Anderson, R., Huchette, J., Tjandraatmadja, C., and Vielma, J. P. Strong convex relaxations and mixedinteger programming formulations for trained neural networks. arXiv preprint arXiv:1811.01988, 2018. *** After Author Response *** After reading the author responses and other reviews, I have decided to maintain my score of 6. I appreciate that the authors tried to run their method on one more standard benchmark from Wong et. al. However, there is not too much I can conclude from 100 out of 10000 test images being verified. I hope that if the paper is accepted, that the authors use their method on the benchmark architecture for all 10000 test images (or at least 1000 of them - 100 is too small). Also, it would be good to report how long each certification procedure took on these larger architectures too.

[Author Response · NeurIPS 2019]

We thank the reviewers for their constructive feedback. We will incorporate the suggestions and improve the overall presentation. We first answer common questions followed by individual ones.

**Q.** *How does k-ReLU and kPoly compare against other methods in terms of precision and timing?*
**R.** k-ReLU allows for tighter relaxations than [1-5] which are based on 1-ReLU. As Rev. 2 points out, k-ReLU is generic and can be combined with [1-5] to improve their precision. For example, eq. (7) in [2] can be solved using more precise relaxations from k-ReLU. In the paper, *kPoly* combines k-ReLU with *DeepPoly*. Experimentally: (i) we show that *kPoly* is more precise than *DeepPoly*, which in turn is as precise or more precise (Sec. 4.2 in [8] and 3.3.4 in [5]) than [1-5] in our experiments, (ii) we compare the timing of *kPoly* with *DeepPoly* (*DeepPoly* has similar timing to [1-3, 5], [4] is much less precise than all of these). Pure MILP based approaches [6-7] do not scale on our benchmarks and time out. Overall, *kPoly* verifies more robustness properties that other verifiers [1-8].

**Q.** *On ConvSmall and ConvBig network, kPoly proves less than state-of-the-art [3,4,7]?*
**R.** The lower provability is due to the networks we used – note that [1-8] on these obtain *lower provability* than *kPoly*. This is because [1-5] are less precise than [8] ([6,7] time out) and [8] is less precise than *kPoly*. We now also ran *kPoly* on more networks from [2] and DiffAI (for 100 images). *kPoly* obtains state-of-the-art provability on both: MNIST ConvBig from DiffAI (93% for $\epsilon = 0.3$) and CIFAR10 ConvSmall from [2] (35% for $\epsilon = 0.03$). We will clarify this point better in the paper.

**Q.** *What is the complexity of kPoly? Can kPoly handle larger networks and ResNets?*
**R.** The complexity of *kPoly* depends on the # of refined neurons (which can be tuned) and the complexity of solving the resulting LP formulations. For larger networks, we refine neurons in the fully connected layer which are much fewer compared to the network size. For networks trained to be robust, the LP formulations are easier to solve, see Table 3: *kPoly* runs slower on the smaller ConvSmall (not trained to be robust) than on the (robustly trained) ConvBig which is $\approx$ 10x larger. *kPoly* scales to ResNets: we now ran *kPoly* on the 100K CIFAR10 ResNet from [2] obtaining 26% provability (100 images) for $\epsilon = 0.03$ ([2] obtains $\approx$22% on full test set). We will include this result.

**Reviewer 1:**
**Q.** *Where are the results on CIFAR10 in the paper?*
**R.** The last network in Tables 2 and 3 is a CIFAR10 network (we will fix this typo).
**Q.** *The authors only report the time limit used for the LP and MILP solvers in their framework.*
**R.** The timings reported in Table 3 are end-to-end timings for the overall verification procedure.

**Reviewer 2:**
**Q.** *Why does* DeepPoly *help?*
**R.** *DeepPoly* helps in two ways: (i) it computes precise initial interval bounds which help the LP solver refine faster, and (ii) it computes precise octagonal bounds that speed-up *kPoly*.
**Q.** *In Equation 8, you need to use $\cap$ instead of $\cup$.*
**R.** No, we first compute convex hulls for each set of $k$-neurons which are then intersected.
**Q.** *On line 151, it is not "$2^n$ convex hulls", but one convex hull that requires $2^n$ inequalities.*
**R.** No, there are $2^n$ polyhedra corresponding to either +ve ($x_i \geq 0$) or -ve ($x_i \leq 0$) value of $x_i$.
**Q.** *On line 147, each $\mathcal{S}$ is actually a subset of $\mathbb{R}^{h x h}$ and not (as is stated) $\mathbb{R}^h$, right?*
**R.** No, each $\mathcal{S}$ is defined over the $h$ neurons in the layer so it is a subset of $\mathbb{R}^h$.
**Q.** *Is the heuristic for selecting the partitioning $\mathcal{I}$ more effective than random partitioning?*
**R.** We ran *kPoly* with 2-ReLU on the MNIST $6 \times 100$ network with $\epsilon = 0.026$. The results show that in most cases, the computed widths of the correct label are smaller with our heuristic.

**Reviewer 3:**
**Q.** *What does precision mean? I think it means the % of images certified to be robust.*
**R.** This is correct, we will fix this.
**Q.** Can you elaborate on *[6], which looks related?*
**R.** Yes, while pure MILP formulations [6-7] do not scale on our benchmarks, we believe *kPoly* may benefit from tighter MILP formulations from [6]. We believe this is an interesting future direction.
**Q.** *Could you explain how the expression in sec. 3.3 leads to the constraints 2-ReLU uses in Fig. 2?*
**R.** Yes, we will explain this and add a detailed derivation in the appendix.
**Q.** *Line 107 - Is 1-ReLU equivalent to the LP-based formulation of FastLin [5] and [3]?*
**R.** No, both Fast-Lin and [3] use a relaxation of 1-ReLU and are thus less precise.
**Q.** *Figure 3 - Why are there two figures? What are (a) and (b)?*
**R.** *DeepPoly* uses either of the 2 relaxations of 1-ReLU in Fig. (a) and (b) based on the input bounds.

**References** 1. Zhang et al. NeurIPS 2018, 2. Wong et al. NeurIPS 2018, 3. Wong et al. ICML 2018, 4. Gowal et al. arxiv 2018, 5. Weng et al. ICML 2018, 6. Anderson et al. IPCO 2019, 7. Tjeng et al. ICLR 2019, 8. Singh et al. ICLR 2019.



[Meta-Review · NeurIPS 2019]

All reviewers were leaning towards acceptance. Unfortunately, in the discussion after the rebuttal it became clear that crucial parts of the paper could not be properly understood e.g. the set S in line 147 is a union of polyhedra whereas it seems that this should be an intersection. Moreover, the notation introduced by the authors (box cap to mean convex hull) was not helpful either. The evaluation is not very helpful as the authors evaluate mainly on non-robust models, whereas the gain on the only robust model (ConvBig) on MNIST is marginal and the same is true for CIFAR-10. It is thus hard to judge how significant the impact of the improved relaxation is for the verification of robust models. On the other hand the reviewers appreciated the idea of k-ReLU relaxation as it can also be used in other verification frameworks. In total this is a borderline paper which requires some improvements in the presentation as well as in the evaluation to judge the paper better.